# High-risk fertility behaviours among women of reproductive ages in the Democratic Republic of the Congo: Prevalence, correlates, and spatial distribution

Zacharie Tsala Dimbuene[1]*, Zemenu Tadesse Tessema[2], Soziac Elise Wang Sonne[3]

1 School of Population and Development Sciences, University of Kinshasa, Kinshasa, Democratic Republic of the Congo, 2 Department of Epidemiology and Biostatistics, University of Gondar, Gondar, Ethiopia, 3 World Bank Africa, Washington, DC, United States of America

☯ These authors contributed equally to this work.
* zacharie.tsala.dimbuene@gmail.com, zacharie.dimbuene@unikin.ac.cd

**Data Availability Statement:** The data underlying the results presented in the study are available

## Abstract

### Background

High-risk fertility behaviour remains a major public health in the Democratic Republic of the Congo, like other sub-Saharan Africa countries, especially because Total Fertility Rate (TFR) is very high in the country, estimated at 6.6 children. Despite the commendable progress in reducing maternal deaths in the region, sub-Saharan Africa is still lagging very behind compared with other regions. Yet, high-risk fertility behaviours are pivotal to improve maternal and child health. This study aims to assess geographical variations of, and to identify risk factors associated with high-risk fertility behaviours among married women in the Democratic Republic of the Congo using the 2013–14 Demographic and Health Survey.

### Methods

Overall, 11,497 married women were selected from a nationally representative using a two-stage sampling design. Standard logistic regressions were performed to identify individual- and household/community-level factors associated with high-risk fertility behaviours. Additionally, interactions between women's age and education, and urban residence were tested. Bernoulli based spatial scan statistics were used to identify the presence of high-risk fertility behaviours spatial clusters using Kulldorff's SaTScan version 9.6 software. ArcGIS 10.7 was used to visualize the spatial variations of high-risk fertility behaviours. Geographically weighted regression (GWR) analysis was employed using Multiscale GWR version 2.0 software.

### Results

Findings indicated that more than two-third of married women exhibited high-risk fertility behaviours in the Democratic Republic of the Congo. Multivariate logistic regression

from https://dhsprogram.com/methodology/survey/survey-display-421.cfm.

**Funding:** The author(s) received no specific funding for this work.

**Competing interests:** The authors have declared that no competing interests exist.

showed that education was negatively and significantly associated with the odds of high-risk fertility behaviours. In contrast, women's age significantly increased the odds of high-risk fertility behaviours. Interactions between urban residence and women's education and age confirmed the urban advantage identified from previous studies. Finally, high-risk fertility behaviours were highly clustered in the Northeastern provinces of the country.

## Conclusion

The study showed that there were significant geographical variations of high-risk fertility behaviours across provinces in the Democratic Republic of the Congo. The paper also identified significant-high hot spots of high-risk fertility behaviours in the Northeastern provinces of the country. To reduce high-risk fertility behaviours, and ultimately improve maternal and child outcomes in the country, policymakers and health planners need to strategically address these inequalities. Finally, this paper highlighted the persistent needs of country-specific studies due to differences across sub-Saharan African countries in terms of social development and cultures.

## Introduction

The United Nations (UN) estimations show that over the last few decades, fertility has declined worldwide; however, the total fertility rate (TFR) is still higher in sub-Saharan Africa (SSA) compared to the rest of the world [1,2]. Indeed, TFR in SSA was estimated at 4.7 births per woman in 2015–2020. This figure is twice than any other region in the world, and therefore it puts women of reproductive ages at higher risks of maternal mortality [1,3–5] while increasing the risks of newborns to die in infancy or early childhood [4]. For instance, maternal mortality ratio (MMR) is estimated at 546 maternal deaths per 100,000 live births in sub-Saharan Africa [5] and it is the region of the world experiencing the lowest decline of MMR [6]. Therefore, not only SSA is putting women and children at higher risks of deceasing, SSA countries did not reach Millennium Development Goals (MDGs) and with the current trends, these countries will not reach the Sustainable Development Goals (MDGs) set out in 2015 by the United Nations. Overall, women of reproductive ages and their children are at very high risk of dying, especially because of the high-risk fertility behaviours (hereafter, HRFB) which is a set of conditions surrounding pregnancies and deliveries, including age at birth, birth spacing, and birth order [7]. Besides the rationales mentioned above emphasizing the "why" to dig into factors associated with high-risk fertility behaviours, it is worthy to mention that HRFB is a major public health concern in SSA which needs to be adequately addressed if SSA countries want to reach SDGs regarding maternal and child mortality. Unmeet family planning needs, child marriage, and weak health systems in most SSA countries are among others, structural factors that still put women and children in jeopardy.

Previous research has identified two set of factors associated with HRFB in developing countries and SSA in particular, including sociodemographics and reproductive health behaviours [3–5,7–12]. Sociodemographics include mother's age, education, marital status, place of residence, among others. Among reproductive health behaviours, studies indicated that history of birth death, antenatal care, facility delivery, and family planning utilization can affect HRFB [7].

This paper examines the correlates of high-risk fertility behaviours in the Democratic Republic of the Congo (DRC) using secondary data analyses of the 2013–2014 Demographic

and Health Survey (DHS) with a special attention to geographical distribution of the outcome. It contributes to the existing body of literature in providing a clear landscape of the high-risk fertility behaviours in a country devastated by successive wars and armed conflict over three decades.

## Materials and methods

### Study setting

The Democratic Republic of the Congo is the biggest Francophone country if SSA located in Central Africa and large enough of 2,345,410 kilometers squares with an estimated population of 92.5 million inhabitants in 2020 [13]. There are justifiable reasons to study high-risk fertility behaviours in this country. First, the country has been devastated by successive wars and armed conflict since 1996. Consequently, the health system has worsened, especially in the Eastern region of the country. This has certainly increased the prevalence of high-risk fertility behaviours in the country. In addition, a sizeable proportion of displaced women are exposed to unfavourable life conditions which increase the risks of mortality and morbidity. Empirically, previous research showed that armed conflict is positively associated with maternal mortality [14]. Second, the DRC is still experiencing high fertility levels. Indeed, the TFR is estimated at 6.6 children per woman [15]. It is among the highest fertility levels worldwide. Indeed, the DRC is ranked at the third place in fertility levels after Niger and Somalia. As such, women are exposed to high-risk fertility behaviours; therefore, putting women and children at higher risks of mortality and morbidity. A study showed that poor institutions in Central Africa are detrimental for children and women [16]. Third, it is expected geographical variations of the distribution of high-risk fertility behaviours given the concentration of health system in urban areas. Yet, fertility is higher in rural areas where existing health conditions can be worse than urban areas, implying that rural women are at higher odds of high-risk fertility behaviours.

### Data

The paper utilizes data from the Demographic and Health Survey (DHS) conducted in the Democratic Republic of the Congo in 2013–2014 (DRC-DHS 2013–14). This is a nationally representative survey, using a two-stage sampling design, which collected information on households, women and men of reproductive ages, anthropometric measures, contraception and family planning, among others [15]. The first stage involved the selection of sample points or clusters from an updated master sampling frame constructed in accordance with DRC's administrative division in 26 provinces or domains. These domains were further stratified into urban and rural areas. From the urban areas, neighbourhoods were sampled from cities and towns whereas for rural areas villages and chiefdoms were sampled. The clusters were selected using systematic sampling with probability proportional to size. Household listing was then conducted in all the selected clusters to provide a complete sampling frame for the second stage selection of households.

   The second stage of selection involved the systematic sampling of the households listed in each cluster, and households to be included in the survey were randomly selected. The rationale for the second stage selection was to ensure adequate numbers of completed individual interviews to provide estimates for key indicators with an acceptable precision. All men and women aged 15–59 and 15–49, respectively, in the selected households were eligible to participate in the survey if they were either usual residents of the household or visitors present in the household on the night before the survey. This paper reports on findings from women individual record file to construct the outcome and independent variables. Analyses are restricted to

married women with at least one live birth and de jure residents, resulting in a final sample of 11,497 women of reproductive ages.

## Ethics statement

Ethical approvals were obtained from the national ethics committee in the Democratic Republic of the Congo before the survey was conducted. Written informed consent was obtained from every participant before they were allowed to participate in the survey. The DHS Program, USA, granted the authors permission to use the data. Since the data were completely anonymous, the authors did not seek further ethical clearance for this study.

## Measures

**Outcome variable.** This paper focuses on high-risk fertility behaviours among married women of reproductive ages. High-risk fertility behaviours were not directly captured in the DHS data; it has been defined elsewhere following DHS guidelines [8–10,17]. In this paper, three criteria are used to build the outcome variable, including age at last birth, birth interval, and birth order. A woman falls in high-risk fertility behaviour category under the following conditions: (*i*) age at birth under 18 years or (*ii*) above 34 years; (*iii*) parity 4 and above; (*iv*) birth spacing is less than 24 months. The presence of any of these conditions puts the woman in a high-risk fertility behaviour category. Readers should note that there are variants in the definition and conceptualization of the high-risk fertility behaviours in the literature. Some scholars have distinguished between one risk and multiple risks to capture and modeling high-risk fertility behaviours [7,10,17,18]. Although this distinction is worthy, this paper has adopted a dichotomization of the variable based on preliminary analyses performed on the dataset and the resulting distribution of the outcome. Indeed, multiple high-risk fertility behaviour categories contained only a few cases and might lead to unstable statistical models. Therefore, the outcome variable is coded 1 if a woman has one or more HRFB-related conditions, and 0 otherwise.

**Independent variables.** Previous literature have identified a number of factors associated with high-risk fertility behaviours in sub-Saharan Africa [4,5,7,11,12,19]. In this paper, correlates of high-risk fertility behaviours were grouped into two categories, including individual-level and household/community-level variables. The first set of variables include respondent's age at survey, education (in completed years), religion, work experience, exposure to media, fertility preferences, current contraception use, antenatal care visits, health insurance, and partner's education. The second set of variables consists of sex of household head, wealth index, community literacy level, community poverty status, urban residence, and the actual province of residence.

## Analytical strategy

In this paper, analyses were performed in three steps. The first step consists of the data cleaning and check of missingness using STATA 15. Essentially, the complete cases (CC) strategy was used for the sake of presentation and to keep the same number of cases for all variables. However, and given the complex survey design (CSD), all the observations in the dataset were retained to ensure that standard errors and confidence intervals are unbiased. In practice, the subpop option of the svy routine was used to ensure that analyses are performed on the subpopulation while restoring the representativeness of the population. Additionally, multicollinearity was checked at this stage. When many covariates are introduced in the model, it is important to check for correlations to avoid wider confidence intervals that result in unreliable probabilities. This was achieved by running a pseudo-ordinary least squares (OLS) model with

the outcome and all independent variables and checking for the variance inflation factors (VIF). The rule of thumb is that a VIF above 5 indicates high correlation and it is a sign of multicollinearity. Findings indicated no multicollinearity issues in the data since the VIF ranged from 1.01 to 2.77.

The second step consists of standard logistic regression using high-risk fertility behaviours as dependent variable and the two set of independent variables. Three models were performed including (*i*) individual-level variables; (*ii*) household/community-level variables and (*iii*) a full model including individual and household/community-level variables. Three interactions were also tested in step 2. First, there are good reasons to think that there is an interaction between mother's education and urban residence. Women's education is significantly and negatively associated with high-risk fertility behaviours [4,10]. Previous studies showed that women's education is higher in urban areas compared with rural areas [20,21]. Second, the paper tested the interaction between current women's age and urban residence because at equal age, fertility might be higher in rural areas compared with urban areas, and therefore putting rural women at higher risks than their urban counterparts. Finally, it is expected that women's age and education might conversely covary since younger generations are likely more educated than older generations.

The third step provides a visualization of the outcome through spatial analyses. The weighted frequency of outcome variable with cluster number and geographic coordinate data was merged in STATA 15. This data was exported to excel then imported to ArcGIS 10.6 for spatial analyses using several techniques briefly described below.

**Spatial autocorrelation.** The spatial autocorrelation (Global *Moran's Index*) statistic [22] measures whether the patterns of the outcome of interest were dispersed, clustered, or randomly distributed in the DRC. Moran's Index (hereafter *Moran's I*) is a spatial statistic used to measure spatial autocorrelation by taking the entire data set and produce a single value which ranges from -1 to +1. *Moran's I* values close to $-1$ indicate that the disease dispersed, whereas Moran's I close to +1 indicate a clustered disease. Finally, *Moran's I* values equal to zero indicate the disease is randomly distributed. To test the statistical significance, the null hypothesis states that the outcome of interest is randomly distributed; otherwise, there might be a presence of spatial autocorrelation. A statistically significant value of Moran's I ($p < 0.05$) suggests the rejection of the null hypothesis, and this is strong evidence of a spatial autocorrelation.

**Hot spot analysis.** *Getis-Ord GI\** statistics [23,24] were computed to measure how spatial autocorrelation varies over the study location by calculating *GI\* statistic* for each area. Z-score is then computed to determine the statistical significance of clustering, and the *p-value* computed for the significance. Statistical output with high *GI\** indicates "hotspot" whereas low *GI\** means a "cold spot".

**Spatial interpolation.** The spatial interpolation analysis [25] is used to predict the unsampled areas based on the sampled areas. The data was not collected from the whole country; therefore, some areas have missing data on the outcome. Spatial interpolation of high-risk fertility behaviours in the sampled areas was used to predict the information on high-risk fertility behaviours in the non-sampled areas. The ordinary Kriging methods of interpolation was used in this study. Details on the methods are presented elsewhere [26].

**Spatial SaTScan analysis.** Spatial scan statistical analysis based on a Bernoulli model was employed to test the presence of statistically significant spatial clusters of high-risk fertility behaviours using Kuldorff's SaTScan version 9.6 software. The spatial scan statistic [27] uses a circular scanning window that moves across the study area to identify the primary and secondary clusters of high-risk fertility behaviours in the country based on prior information collected in the DHS.

## Result

### Socio-demographics of the sample

Table 1 presents the characteristics of the sample (*N* = 11, 497) included in the analyses. Over-all, 68.5% of women of reproductive ages in the Democratic Republic of the Congo had at least one high-risk fertility behaviour. This includes age at birth less than 18 years or above 34 years; a birth interval less than 24 months; or birth order above four live births.

Turning to the key independent variables, findings indicated that women were aged of 31.3 years on average. This is also means that, on average, women were close to the risk group (35 + years). The average number of completed year of education was about 5.4 years (*S.D.*: 4 years). Put differently, one might say that women did not complete the first level of the official educational system which last six years. The expansion of religious practices in the country has brought new landscape. Indeed, while most people were either Catholics or Protestants in the past, findings indicated a shift towards the "*Other Christians*" category (40%); this includes the second generation of churches, the so-called "*Églises de reveil*", followed by Protestant and Catholic with 28%. Surprisingly, most women (70%) were working at the time of the survey. The fertility preferences mimicked the higher fertility level observed in the country with an average ideal number of children estimated at 6.6 (*S.D.*: 2.7).

The decision to use contraception could also be an indication to fall under risk categories. In the sample, a small percentage of married women (18%) chose to use contraception them-selves or jointly with their husband/partner. This also means that in most cases, other people interfere in the decision to use contraception, which is not surprising in the context of sub-Saharan Africa. With regards to antenatal care attendance, a sizeable percentage of married women (approximately, 61%) had less than four number of antenatal care visits; yet the World Health Organization (WHO) recommend that a pregnant woman should have at least four ANC visits. Unlike women, most husbands/partners—a total of 69%, reached at least second-ary education. As expected, most women (64%) lived in rural areas. The number of selected women were unequally distributed across provinces. The percentage ranged from 7.2% in North Kivu to 17.7% in Bandundu, which is not the most populous province in the country.

### Factors associated with high-risk fertility behaviours

Table 2 and Figs 1–3 present multivariate findings from logistic regression of having high-risk fertility behaviours on (*i*) individual characteristics of the women (Model 1); (*ii*) household/community characteristics (Model 2); and both individual- and household/community-level characteristics (Model 3). Model 1 showed interesting findings. Indeed, the odds of high-risk fertility behaviours among women of reproductive ages in the DRC increased with women's current age and decreased with the educational level. An addition year of age increased the odds of high-risk behaviour by 29%. In contrast, an additional completed year of education decreased the odds of experiencing high-risk behaviour by 7.9%. Surprisingly, husband/part-ner's education exhibited a positive association with high-risk fertility behaviour even though the coefficients did not reach statistical significance, except for secondary level. Having a hus-band/partner with secondary with secondary education increased by 35% the odds of high-risk fertility behaviour.

Model 2 about household- and community-level characteristics also showed unexpected findings. In fact, living better-off households increased the odds of high-risk fertility behav-iours in the DRC. Specifically, living in middle and rich households increased by 25% and 32% the odds of high-risk fertility behaviours, respectively. In contrast, advantaged neighbourhoods had opposite effects on the odds of high-risk fertility behaviours. Living in advantaged

**Table 1. Descriptive statistics of married women in the Democratic Republic of the Congo.**

| Variables | Mean or % | S.D. |
|---|---|---|
| **Dependent variable** | | |
| High-risk fertility behaviour | 68.5 | |
| **Independent variables** | | |
| **Individual-level characteristics** | | |
| Women's age at survey | 31.3 | 8.2 |
| Women's education (in completed years) | 5.4 | 4.0 |
| Religion | | |
| Catholic | 27.6 | |
| Protestant | 28.4 | |
| Other Christian | 39.8 | |
| Other religions | 4.2 | |
| Is currently working (% YES) | 69.9 | |
| Exposure to media (% YES) | 35.1 | |
| Ideal number of children | 6.6 | 2.7 |
| Decision to use contraception | | |
| Oneself or joint decision with husband/partner | 17.8 | |
| Has health insurance (% YES) | 4.2 | |
| Antenatal care attendance | | |
| No ANC visit | 8.7 | |
| 1–3 ANC visits | 52.0 | |
| 4+ ANC visits | 39.3 | |
| Partner's educational level | | |
| No education | 8.6 | |
| Primary | 22.4 | |
| Secondary | 59.7 | |
| University or higher | 9.3 | |
| **Household- and Community-level characteristics** | | |
| Household Head is Female | 15.2 | |
| Household Wealth Index | | |
| Poor (40%) | 42.5 | |
| Middle (20%) | 20.6 | |
| Rich (40%) | 37.0 | |
| Community Literacy Level | | |
| Low | 30.2 | |
| Medium | 37.0 | |
| High | 32.8 | |
| Community Socioeconomic Status | | |
| Low | 63.8 | |
| High | 36.2 | |
| Urban residence | | |
| Rural | 68.1 | |
| Urban | 31.9 | |
| Province of residence | | |
| Kinshasa | 8.1 | |
| Bandundu | 17.7 | |
| Kongo Central | 4.2 | |
| Equateur | 14.4 | |

(*Continued*)

**Table 1.** (Continued)

| Variables | Mean or % | S.D. |
|---|---|---|
| Kasai Occidental | 7.6 | |
| Kasai Oriental | 11.3 | |
| Katanga | 9.8 | |
| Maniema | 3.4 | |
| North Kivu | 7.2 | |
| Orientale | 9.1 | |
| South Kivu | 7.3 | |
| Sample size (*N*) | 11,497 | |

Source: DHS DRC 2013–14.

neighbourhoods decreased by 23% the odds of high-risk fertility behaviours. Model 3 including individual- and household/community-level characteristics did not change the direction and magnitude of the results observed previously, except that being in "primary education category" reached a marginal statistical significance ($p < 10\%$).

*Testing interactions.* Three two-way interactions were tested in the analyses: women's education and place of residence, women's age and place of residence, and women's current age and education. Findings are presented in Figs 1–3. As previously observed (Model 1), the odds of high-risk fertility decreased with the level of women's education as expected (Fig 1).

As expected, the effects of women's education on the likelihood of high-risk fertility behaviour varied between rural and urban areas. At all levels of women's education, the odds of high-risk fertility behaviours are higher in rural areas compared with urban areas. However, the differences between urban and rural women are not statistically significant when women had completed less than five years of education.

From Fig 2, findings indicated that the odds of high-risk fertility behaviours are higher among rural than urban women at any specific age. The differences increased between ages 20 and 35 and they are statistically significant, illustrating the urban advantage in health issues, including high-risk fertility behaviours. The gap vanished at older ages and became statistically insignificant.

Fig 3 plotted the predicted probabilities in combining educational levels and women's current age. To better capture the wealth of information contained in this graph, one might choose values of both independent variables for a given probability. Previous studies showed that young women can be at higher risks. However, Fig 3 showed that the levels of risks decreased with the level of education.

## Spatial analyses of the high-risk fertility behaviours in the Democratic Republic of the Congo

**Spatial distribution.**   The spatial distribution of high-risk fertility behaviours was performed using ArcGIS version 7.0 software and the results are displayed in Fig 4. A total of 540 Enumeration areas were located and used in this analysis. The proportion of high-risk fertility behaviours varies in each enumeration area (cluster). As indicated in the map, it ranged from 21.01% to 89.47%. The red color indicated high percentage of high-risk fertility behaviours and the green color indicated low percentage of high-risk fertility behaviours (Fig 4).

**Spatial autocorrelation.**   The spatial autocorrelation result revealed that there is clustering effect in the high-risk fertility behaviours among reproductive age women in the DRC. The

**Table 2. Adjusted Odd Ratio (AOR) of experiencing high-risk fertility behaviours among married women in the Democratic Republic of the Congo.**

| VARIABLES | Model 1 | Model 2 | Model 3 |
|---|---|---|---|
| **Individual-level characteristics** | | | |
| Women's current age | 1.285*** | | 1.295*** |
| | (1.269–1.301) | | (1.278–1.312) |
| Woman's education (in completed years) | 0.921*** | | 0.927*** |
| | (0.900–0.944) | | (0.903–0.952) |
| Religion (Ref.: Catholic) | | | |
| Protestant | 1.039 | | 0.979 |
| | (0.855–1.264) | | (0.804–1.193) |
| Other Christians | 1.187* | | 1.314*** |
| | (0.983–1.435) | | (1.087–1.589) |
| Other religions | 1.037 | | 1.057 |
| | (0.743–1.445) | | (0.747–1.494) |
| Is currently working (Ref.: Unemployed) | 1.002 | | 0.998 |
| | (0.866–1.159) | | (0.857–1.162) |
| Exposure to media (Ref.: No exposure) | 1.133 | | 1.054 |
| | (0.971–1.322) | | (0.895–1.243) |
| Ideal number of children | 1.063*** | | 1.049*** |
| | (1.020–1.107) | | (1.014–1.085) |
| Decision to use contraception (Ref.: Other people) | | | |
| Myself or joint decision with my husband/partner | 0.930 | | 0.976 |
| | (0.763–1.134) | | (0.801–1.188) |
| Antenatal attendance (Ref.: No ANC visit) | | | |
| 1–3 ANC visits | 0.960 | | 0.901 |
| | (0.770–1.197) | | (0.731–1.112) |
| 4+ ANC visits | 0.890 | | 0.833* |
| | (0.716–1.106) | | (0.672–1.033) |
| Has health insurance (Ref.: No) | 1.141 | | 0.973 |
| | (0.764–1.704) | | (0.580–1.632) |
| Partner's educational level (Ref.: No education) | | | |
| Primary | 1.252 | | 1.370* |
| | (0.921–1.702) | | (0.992–1.890) |
| Secondary | 1.347** | | 1.422** |
| | (1.042–1.740) | | (1.083–1.867) |
| University or higher | 1.104 | | 1.173 |
| | (0.725–1.681) | | (0.758–1.817) |
| **Household- and Community-level characteristics** | | | |
| Household Head is Female (Ref.: Male) | | 1.126 | 1.155 |
| | | (0.969–1.309) | (0.929–1.436) |
| Household Wealth Index (Ref.: 40% Poor) | | | |
| Middle (20%) | | 1.250*** | 1.289** |
| | | (1.060–1.475) | (1.061–1.566) |
| Rich (40%) | | 1.317*** | 1.190 |
| | | (1.115–1.555) | (0.922–1.535) |
| Community Literacy Level (Ref.: Low-1st tercile) | | | |
| Medium (2nd tercile) | | 1.020 | 1.251** |
| | | (0.888–1.172) | (1.006–1.556) |
| High (3rd tercile) | | 0.770*** | 0.965 |

(*Continued*)

**Table 2.** (Continued)

| VARIABLES | Model 1 | Model 2 | Model 3 |
|---|---|---|---|
| | | (0.650–0.913) | (0.738–1.263) |
| Community Socioeconomic Status (Ref.: 1$^{st}$ half) | | | |
| High (2$^{nd}$ half) | | 1.081 | 1.249* |
| | | (0.928–1.260) | (0.995–1.569) |
| Urban residence (Ref.: Rural) | | 0.883 | 0.869 |
| | | (0.737–1.058) | (0.673–1.122) |
| Province of residence (Ref.: Kinshasa) | | | |
| Bandundu | | 0.868 | 1.132 |
| | | (0.674–1.116) | (0.817–1.568) |
| Kongo Central | | 0.739 | 0.876 |
| | | (0.466–1.172) | (0.497–1.544) |
| Equateur | | 0.977 | 1.572** |
| | | (0.747–1.278) | (1.107–2.231) |
| Kasai Occidental | | 0.901 | 1.322 |
| | | (0.676–1.200) | (0.894–1.955) |
| Kasai Occidental | | 0.782* | 1.224 |
| | | (0.601–1.019) | (0.827–1.813) |
| Katanga | | 1.052 | 1.929*** |
| | | (0.793–1.394) | (1.396–2.666) |
| Maniema | | 0.841 | 1.385 |
| | | (0.624–1.133) | (0.904–2.121) |
| North Kivu | | 1.049 | 1.809*** |
| | | (0.733–1.502) | (1.182–2.770) |
| Orientale | | 0.725** | 1.148 |
| | | (0.557–0.945) | (0.818–1.611) |
| South Kivu | | 1.197 | 3.009*** |
| | | (0.911–1.572) | (1.968–4.600) |
| Observations | 11,497 | 11,497 | 11,497 |

95% Confidence interval of Adjusted Odd Ratio (AOR) in parentheses.

Significance level

*** p<0.01

** p<0.05

* p<0.1.

Source: DHS DRC 2013–14.

top left corner of Fig 5 reports inferential statistics, including Moran's *Index*, *z*-score and *p*-value whether there is a statistically clustering effect of the outcome in the data. From these statistics, there is clearly a clustering effect in the data. The z-score of 3.62 (p-value = 0.000297) indicated that the clustering of HRFB observed in the DRC is not *random*. The result showed that the observed value is greater than the expected value and the *p-value* is < 0.001, it is statistically significant. The clustered patterns (on the right sides) show high rates of high-risk fertility behaviours occurred in the study setting. The outputs have automatically generated keys on right and left sides of each panel. The bright red and blue colors to the end tails indicate increased significance level.

**Hot spot analysis.** The hot spot analyses identified both hot (areas which had high proportion high-risk fertility behaviours located by red colors) and cold spots (areas which had

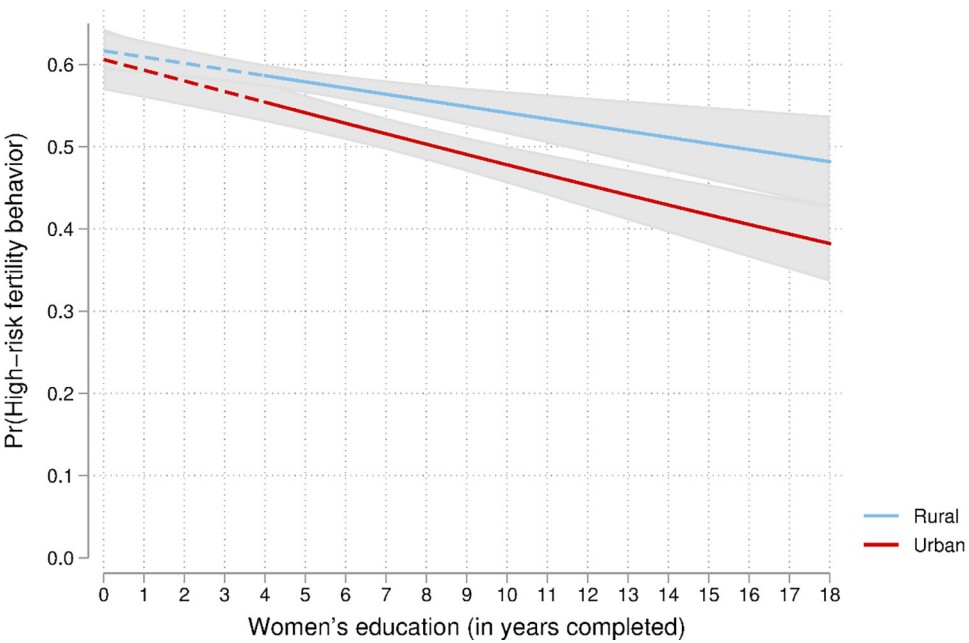

**Fig 1. Predicted probability of high-risk fertility behaviour by women's education and place of residence among married women in the Democratic Republic of the Congo.** Note: Group difference (rural vs urban) is significant ($p < 0.05$) when lines are solid.

low proportion high-risk fertility behaviours located by blue colors) areas. The hot spot areas were concentrated in four provinces, including North Kivu and South Kivu, Equateur, and Katanga. The cold spot areas were located in Bandundu, Kongo-Central, and Kinshasa (Fig 6).

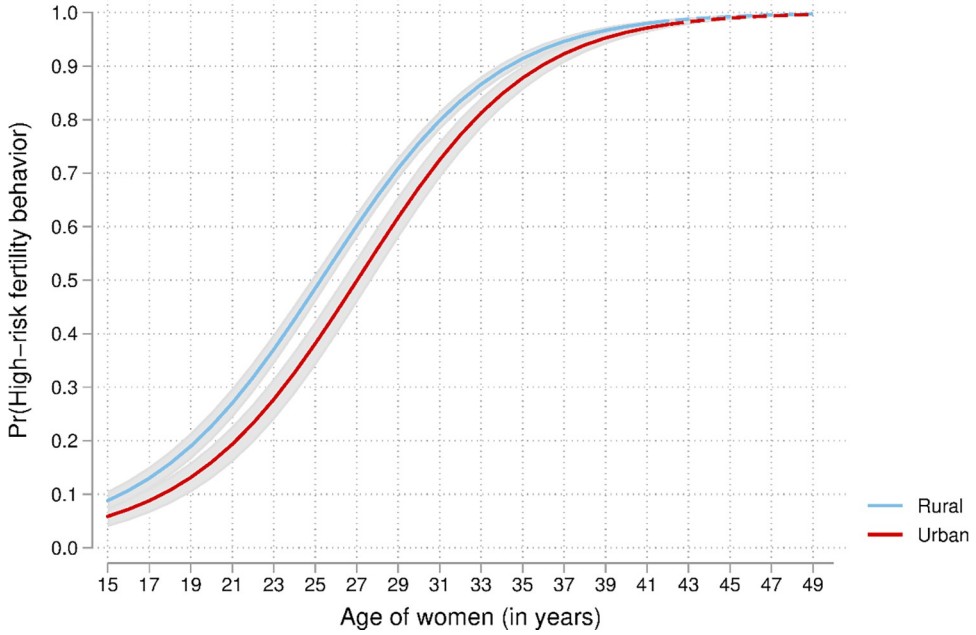

**Fig 2. Predicted probability of high-risk fertility behaviour by age and place of residence among married women in the Democratic Republic of the Congo.** Note: Group difference (rural vs urban) is significant ($p < 0.05$) when lines are solid.

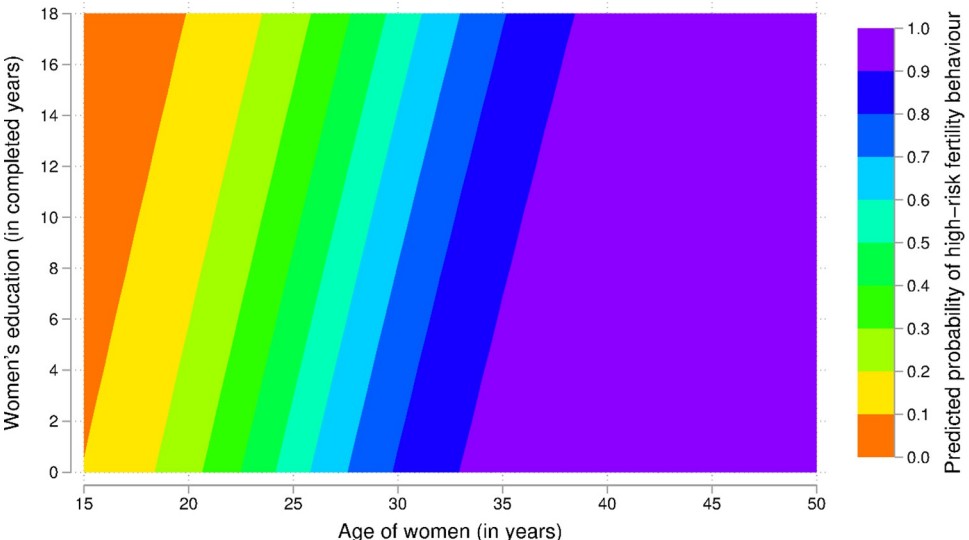

**Fig 3. Predicted probability of high-risk fertility behaviour by age and education among married women in the Democratic Republic of the Congo.**

**Spatial interpolation of high-risk fertility behaviours.** The interpolation analysis results showed that the high predicted percentage of high-risk fertility behaviours was found in Equateur, Orientale, South Kivu, Maniema, and Katanga. In contrast, low predicted percentage of

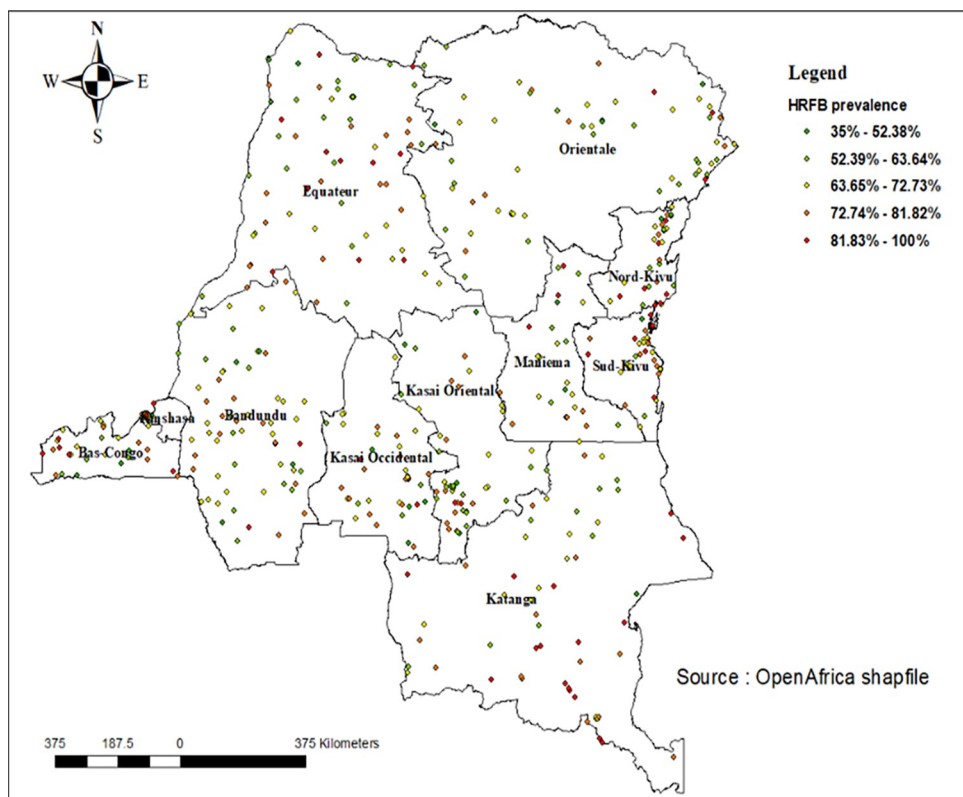

**Fig 4. Spatial distribution of high-risk fertility behaviours among married women in the Democratic Republic of the Congo.**

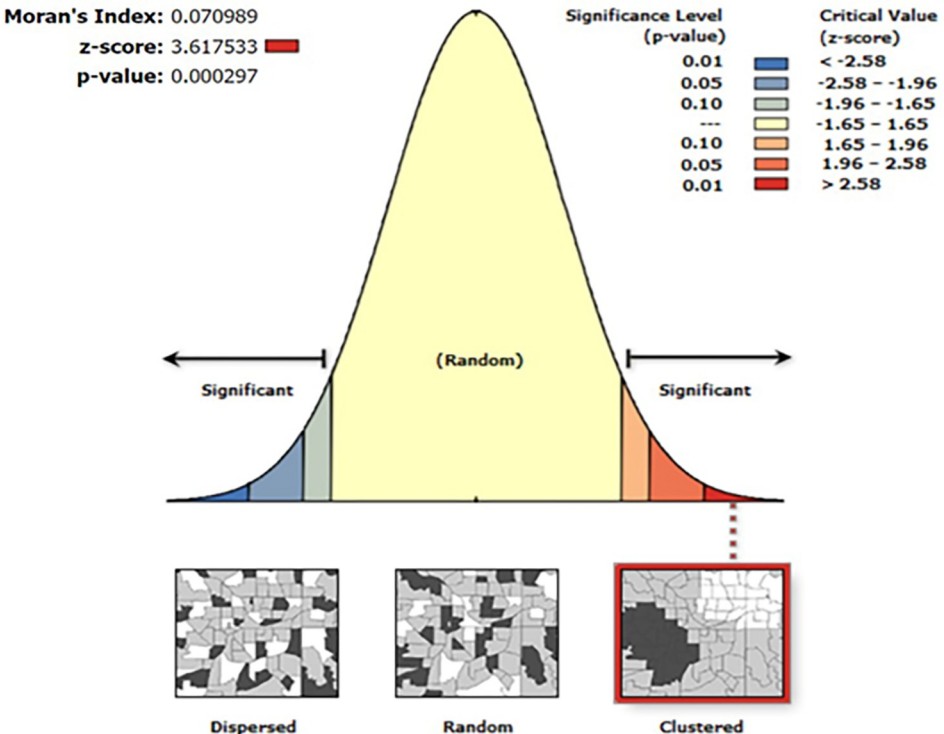

**Fig 5. Spatial autocorrelation result of high-risk fertility behaviours among married women in the Democratic Republic of the Congo.**

high-risk fertility behaviours were observed in Bandundu, Kongo-Central, and Kinshasa (Fig 7).

**SaTScan analyses.** Most likely (primary clusters) and secondary clusters of high-risk fertility behaviours were identified. A total of 492 significant clusters were identified. Of which, 8 of them were most likely (primary) clusters and 34 was a secondary cluster. The primary clusters spatial window is found in the Northeastern part of the country and is located at (10.859151 S, 26.635813 E) / 123.10 km with $p < 0.001$. It showed that women of reproductive ages within the spatial window had 1.19 times higher odds of experiencing high-risk fertility behaviours than women outside the window. The secondary cluster window was located Bas-Congo and Kinshasa at (2.838276 S, 28.804380 E) / 172.25 km (Table 3, Fig 8). The result revealed that women within the spatial window had 1.07 times higher odds of experiencing high-risk fertility behaviours than women outside the window.

## Discussion

This paper investigated the prevalence of, and factors associated with high-risk fertility behaviours in the Democratic Republic of the Congo. Previous research showed that high-risk fertility behaviours are associated with adverse maternal and child outcomes [11]. Findings indicated that 69% of women of reproductive ages in the country exhibited high-risk fertility behaviours. Significant variations were observed across provinces regarding the geographical distribution of the high-risk behaviours in the country. Indeed, the percentage of high-risk behaviours ranged from 63% to 75% in the province Orientale and South Kivu, respectively. These two provinces are in the Northeastern part of the country where recurrent wars and armed conflict have been devastating in the last 20 years. In sum, the percentage of women of

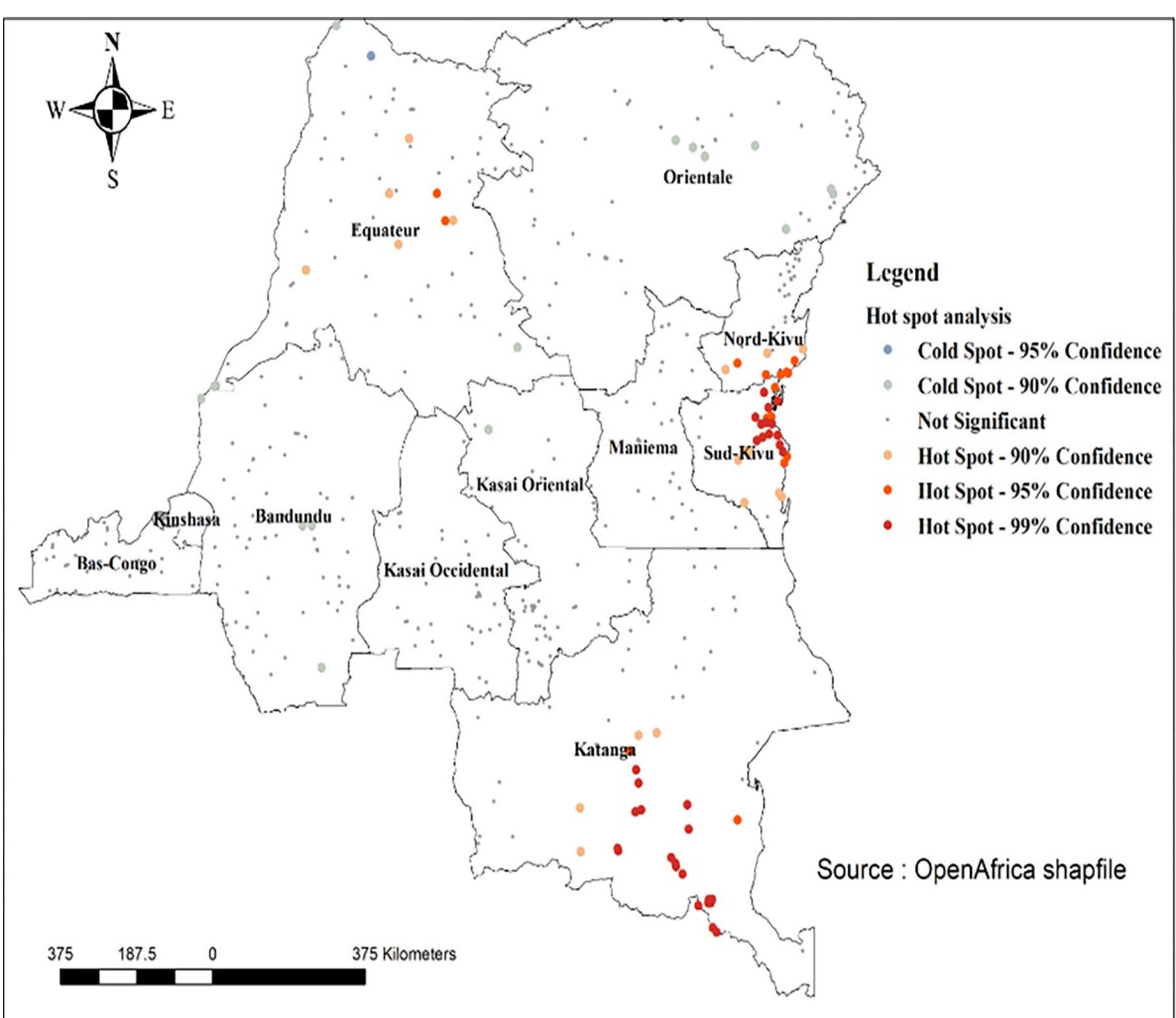

**Fig 6. Hot spot results of high-risk fertility behaviours among married women in the Democratic Republic of the Congo.**

reproductive ages in the DRC with high-fertility behaviours is in line with previous research [5,7].

Women's current age and education were among significant correlates of high-risk fertility behaviours at individual-level. It is worthy to mention that most studies which included these two variables in their analyses as categorical variables. This study adopted a totally different approach and included them as continuous variables. This is very important from a programmatic perspective because policymakers can easily understand the added value to improve women' education on one hand, and on the other hand, the benefit of educating women to give birth within a lower risk bracket. In fact, women's age increased the odds of high-risk fertility behaviour while women's education decreased the odds of high-risk fertility behaviours. This is in line with previous studies which found that the probability of high-risk fertility behaviours decreased as women's education increased [11]. Husband/partner's educational level showed unexpected findings. While previous research reported that husband/partner's education is associated with lower odds of high-risk fertility behaviours in sub-Saharan Africa [11], this paper found that husband/partner's education increased the probability of high-risk fertility behaviours. However, this study differs from the previous research in two ways. First,

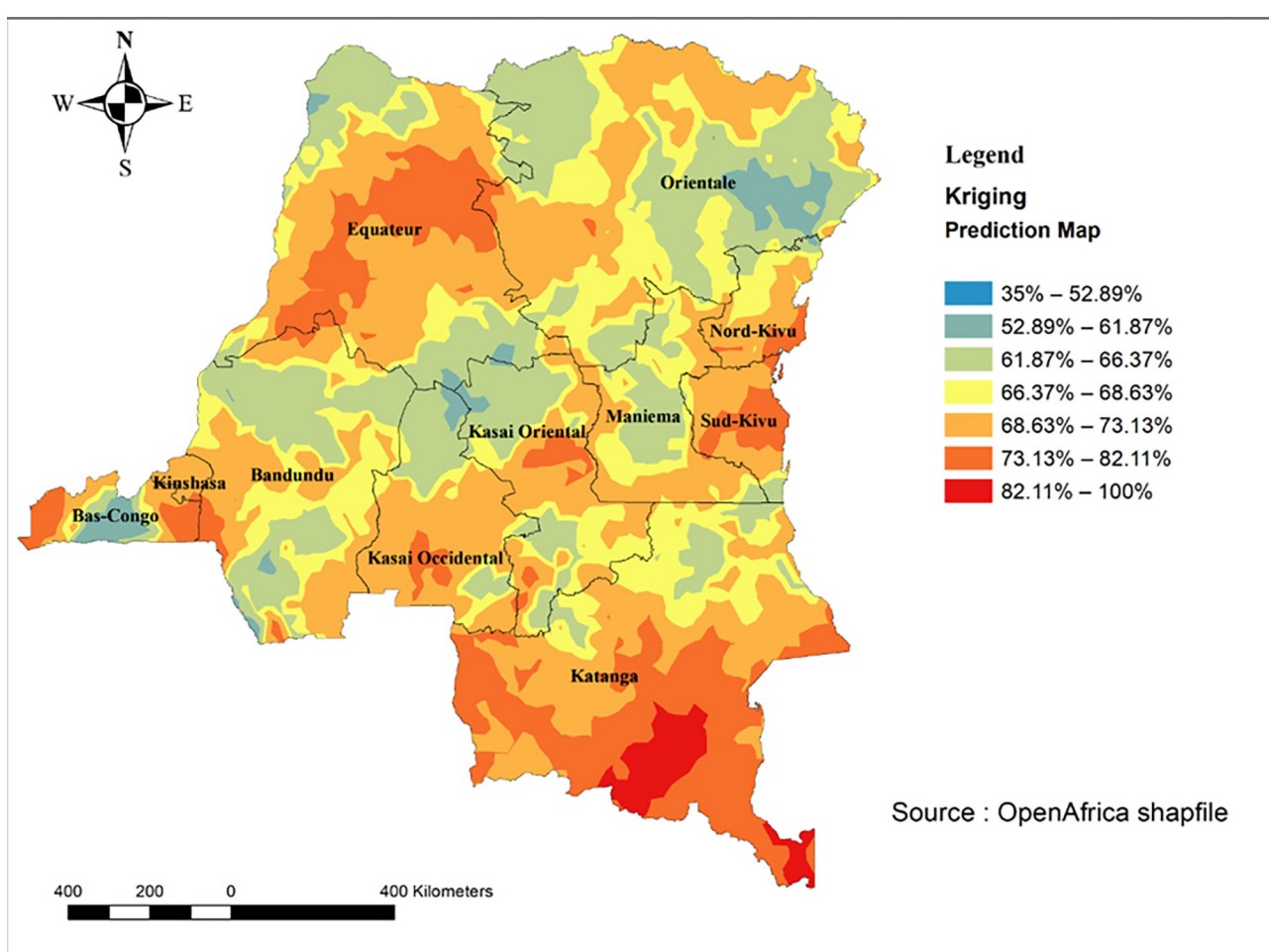

**Fig 7. Spatial interpolation results of high-risk fertility behaviours among married women in the Democratic Republic of the Congo.**

Seidu et al. included data from 27 countries from SSA [11]. These countries are obviously different with regards to socio-economic development and cultures. Therefore, pooling data could mask country-specific context. Second, this study included women of all marital status, in which case the husband/partner's education is meaningless. In the current study, analyses were restricted to married and cohabiting women at the time of the survey. Besides the difference in the scope, there are other plausible explanations that the paper points out to understand how and why husband/partner's education has a positive association with high-risk fertility behaviours in the context of high fertility such the DRC.

It is possible that educated husbands/partners have better socio-economic situation in the country. In a culture which values high fertility under the assumption that children are God's

**Table 3. SaTScan analysis result summary of HRFB among married women in the Democratic Republic of the Congo.**

| Cluster | Enumeration areas (cluster) identified | Coordinate/radius | Population | Case | RR | LLR | *p-value* |
|---------|----------------------------------------|-------------------|------------|------|-----|------|-----------|
| 1 | 8 | (10.859151 S, 26.635813 E) / 123.10 km | 197 | 179 | 1.19 | 13.44 | 0.00014 |
| 2 | 34 | (2.838276 S, 28.804380 E) / 172.25 km | 792 | 39 | 1.07 | 9.02 | 0.98 |

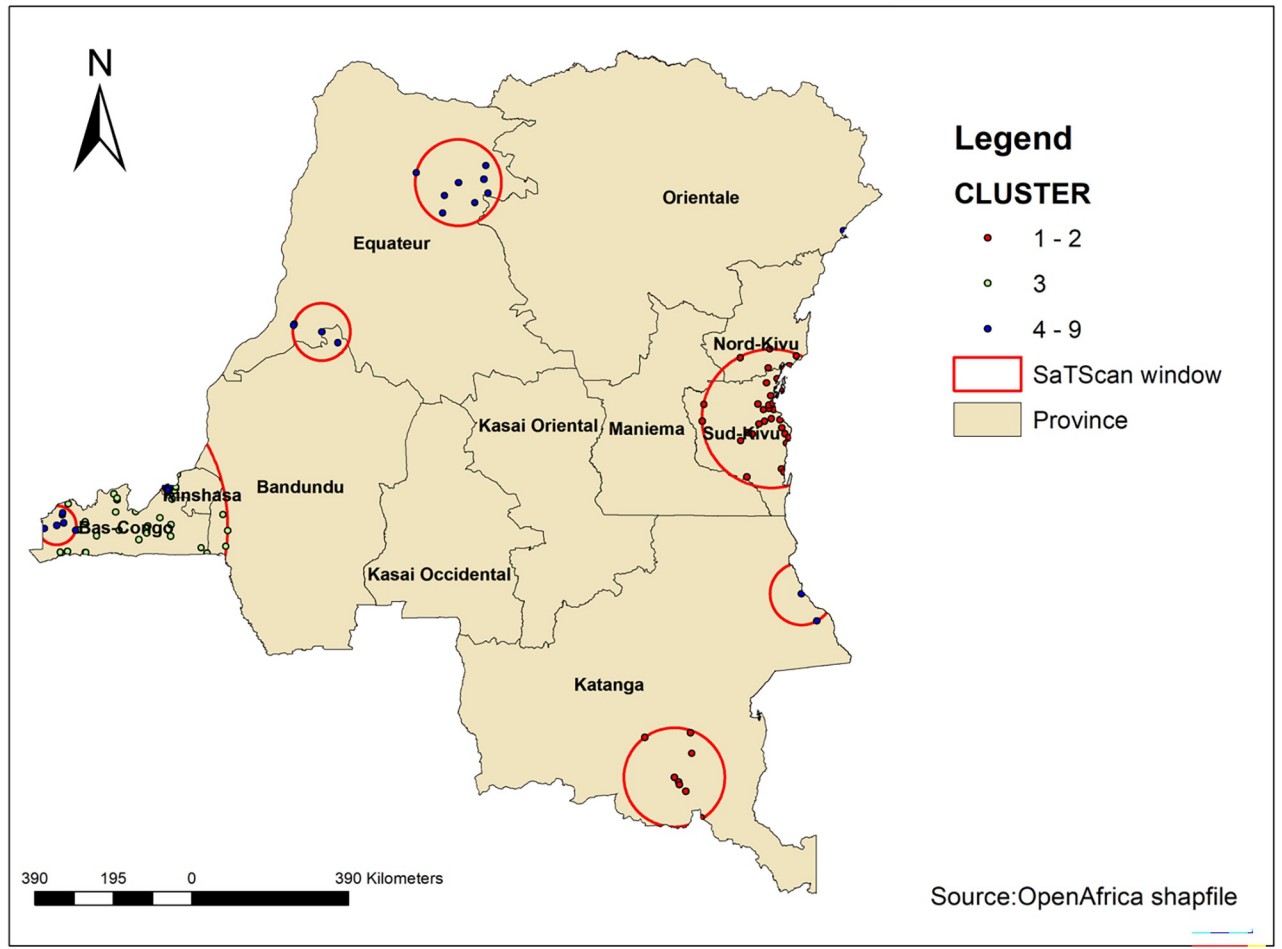

**Fig 8. SaTScan analyses of high-risk fertility behaviours among married women in the Democratic Republic of the Congo.**

blessings, these men tend to have more children than their uneducated counterparts. To test this assumption, an analysis of variance between children ever born and husband/partner's education was performed, and the average number of children ever born, as an indicator of fertility preferences, was computed (*Results not shown*). Findings showed that the differences between the four categories of husband/partner's education were not statistically significant. This finding might explain why husband/partner's education had unexpected association with high-risk fertility behaviours.

The associations between household wealth index (HWI), community literacy level and the high-risk fertility behaviours provided intriguing findings. While previous research reported negative but not significant associations between HWI and high-risk fertility behaviours among women of reproductive ages in Eastern African countries [7] and Ethiopia [5], this study showed that living in better-off households at the time of the survey increased the probability of high-risk fertility behaviours in the DRC. As with the community literacy level, a proxy of social development, better-off neighbourhoods were significantly associated with lower probability of high-risk fertility behaviours. Previous studies reported similar findings even though they did not reach statistical significance [5,7,11].

Spatial analyses revealed appealing findings which have policy and programmatic implications. First, spatial analyses revealed the non-randomness of high-risk fertility behaviours

among married women in the DRC. Such findings mean that effective planning of interventions and programs tackling high-risk fertility behaviours in the DRC should account for the spatial distribution of the outcome. Second, most hot spots were found in the Northeastern provinces in the DRC, including North Kivu and South Kivu, Equateur, and Katanga. The country has been devastated by successive wars and armed country since 1996, and these provinces were the most affected. However, more research is needed to quantify the effects of wars and armed conflict on high-risk fertility behaviours among married women in the DRC. Third, interpolation analyses confirmed the concentration of high-risk fertility behaviours in the Northeastern provinces in the DRC, which red spots indicating higher risks among married women.

This paper contributed to the existing literature by advancing previous analyses. First, the interactions between women's current age, education, and place of residence were fully tested. This study confirmed the urban advantage which is critical if policymakers and stakeholders want to reduce inequalities between urban and rural areas. The analysis of interactions showed that at each age or educational level, the probability of high-risk probability among women of reproductive ages was higher in rural areas compared with urban areas. Women's education played an important role to understand these findings as showed in the paper. Besides women's education, there are other characteristics which are likely important to unpack these differences between rural and urban areas. These include antenatal care (ANC), exposure to media, health insurance, and working status of the women. Urban areas are often more equipped than rural areas concerning ANC utilization, exposure to media and health insurance. However, these factors did not reach statistical significance with high-risk fertility behaviours, and therefore no definite conclusions can be drawn from this study to understand urban advantage.

Second, this is the first study to our knowledge, to focus on spatial variations of high-risk fertility behaviours among women of reproductive ages in the DRC. Yet, improving our knowledge about the spatial distribution of high-risk fertility behaviours is of paramount importance to critically mitigate the differences between urban and rural areas on one hand, and between- and within-province differences. The study also identified hot spots of high-risk fertility behaviours in the country which, if not addressed, will perpetuate high prevalence of HRFB in the country while making women more vulnerable in a context of high fertility and weak health systems. Indeed, spatial analyses showed how the Northeastern part of the country is of great concern with regards to high-risk fertility behaviours.

## Study limitations

The paper used a nationally representative dataset of women of reproductive ages. However, it has some limitations. One of the limitations of the study relies on the cross-sectional nature of the data. Indeed, cross-sectional data can only detect associations between the outcome of interest and putative correlates. Therefore, no definite conclusion can be drawn regarding the causality between high-risk fertility behaviours and the putative factors. Another limitation of the paper is the unobserved effects of wars and armed conflict in a country which has been devastated over the last two decades. This is beyond the scope of a paper using secondary data analysis.

## Conclusion

The paper showed that high-risk fertility behaviours among married women of reproductive ages in the DRC remain a great public health concern in the country given its high prevalence. More than two-third married women included in the analyses exhibited high-risk fertility

behaviours. Additionally, the paper demonstrated the urban advantage: urban women have lower probability of high-risk fertility behaviours compared with their rural counterparts. This finding has programmatic implications if policymakers and stakeholders really want to reduce the inequalities between urban and rural areas. Programs and interventions on women of reproductive ages with a focus on health education and behavioural changes are highly recommended, especially those encouraging contraceptive use and optimal birth spacing. Realistically, the government could legislate and implement interventions on age at first marriage with the ultimate goal of increasing age at first birth while encouraging the utilization of contraception. Finally, there are significant spatial differences with the Northeastern part of the country being more affected.

## Acknowledgments

The authors wish to express their gratitude to the DHS Program, USA, for its generosity in providing them full access to the data. They also wish to acknowledge institutions of the Democratic Republic of the Congo that played critical roles in the data collection process.

## Author Contributions

**Conceptualization:** Zacharie Tsala Dimbuene, Zemenu Tadesse Tessema, Soziac Elise Wang Sonne.

**Data curation:** Zacharie Tsala Dimbuene, Zemenu Tadesse Tessema, Soziac Elise Wang Sonne.

**Formal analysis:** Zacharie Tsala Dimbuene, Zemenu Tadesse Tessema, Soziac Elise Wang Sonne.

**Investigation:** Zacharie Tsala Dimbuene.

**Methodology:** Zacharie Tsala Dimbuene, Zemenu Tadesse Tessema, Soziac Elise Wang Sonne.

**Project administration:** Zacharie Tsala Dimbuene.

**Software:** Zacharie Tsala Dimbuene, Zemenu Tadesse Tessema.

**Supervision:** Zacharie Tsala Dimbuene.

**Validation:** Zacharie Tsala Dimbuene, Zemenu Tadesse Tessema.

**Visualization:** Zacharie Tsala Dimbuene, Zemenu Tadesse Tessema, Soziac Elise Wang Sonne.

**Writing – original draft:** Zacharie Tsala Dimbuene, Zemenu Tadesse Tessema, Soziac Elise Wang Sonne.

**Writing – review & editing:** Zacharie Tsala Dimbuene, Zemenu Tadesse Tessema, Soziac Elise Wang Sonne.

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
