## [Decision Letter · Decision Letter 0]

29 Nov 2022

PONE-D-22-12958

High-risk fertility behaviors among women of reproductive ages in the Democratic Republic of the Congo: Prevalence, Correlates and Spatial distribution

PLOS ONE

Dear Dr. Tsala Dimbuene,

Thank you for submitting your manuscript to PLOS ONE. After careful consideration, we feel that it has merit but does not fully meet PLOS ONE’s publication criteria as it currently stands. Therefore, we invite you to submit a revised version of the manuscript that addresses the points raised during the review process.

We look forward to receiving your revised manuscript.

Kind regards,

Oyelola A. Adegboye, PhD

Academic Editor

PLOS ONE

“NO authors have competing interests”

3. We note that [Figures 4-8] in your submission contain [map/satellite] images which may be copyrighted. All PLOS content is published under the Creative Commons Attribution License (CC BY 4.0), which means that the manuscript, images, and Supporting Information files will be freely available online, and any third party is permitted to access, download, copy, distribute, and use these materials in any way, even commercially, with proper attribution. For these reasons, we cannot publish previously copyrighted maps or satellite images created using proprietary data, such as Google software (Google Maps, Street View, and Earth). For more information, see our copyright guidelines: http://journals.plos.org/plosone/s/licenses-and-copyright.

a. You may seek permission from the original copyright holder of Figures 4-8 to publish the content specifically under the CC BY 4.0 license. 

Natural Earth (public domain): http://www.naturalearthdata.com/.

Reviewers' comments:

Reviewer's Responses to Questions

**Comments to the Author**

1. Is the manuscript technically sound, and do the data support the conclusions?

Reviewer #1: Yes

Reviewer #2: Yes

Reviewer #3: Partly

2. Has the statistical analysis been performed appropriately and rigorously? 

Reviewer #1: Yes

Reviewer #2: Yes

Reviewer #3: Yes

3. Have the authors made all data underlying the findings in their manuscript fully available?

Reviewer #1: Yes

Reviewer #2: No

Reviewer #3: Yes

4. Is the manuscript presented in an intelligible fashion and written in standard English?

Reviewer #1: Yes

Reviewer #2: Yes

Reviewer #3: Yes

5. Review Comments to the Author

Reviewer #1: Thanks to the authors for studying an important issue. We know sub sharan African countries are more vulnerable for public health outcomes for different reasons. I am happy reading the paper it is really good enough and it has value for further investigation in this topic. However I have some addressable comments.

What is the power of the analysis?

Why did you do the study that is missing in the rationale section? I think you did multilevel modelling but it is showing multivariate please clarify it.

The conclusion section doesn’t make any concrete recommendations please specify it.

Please add a recommendations or policy implications section.

Reviewer #2: “High-risk fertility behaviors among women of reproductive ages in the Democratic Republic of the Congo: Prevalebce, Correlates and Spatial distribution”

By Zacharie Tsala Dimbuene, Zemenu Tadesse Tessema, and Soziac Elise Wang Sonne

Reviewed by Bluette Arcady Mongoue, October 2022

This is an important empirical paper that should benefit from wide diffusion; more particularly in the context of Africa south of the Sahara where the number of deaths following high-risk pregnancies or neonatal / perinatal deaths remain greatly high compared to the rest of the world. The study is relevant in such context where health care is still quite precarious; mainly in regions that have witnessed political instability and violence.

As the authors mentioned, the paper is not the first to investigate the correlates and implications of High fertility risk behaviors among women of reproductive age. However, it differs from the existing literature in that it identifies the geospatial distribution of risky fertility decision in the Democratic Republic of the Congo. This work could therefore constitute a good starting point for the Congolese authorities who would like to limit risky fertility behaviors among Congolese women.

The paper identifies factors that are correlated to women risky fertility behaviors in RDC.

The authors conclude that women’s current age significantly increased the odds of high-risk fertility behaviors. As mentioned in the pater, they obtained this result by performing a standard logistic regression of an indicator of high-risk fertility behaviors on age education and other individuals and household characteristics. That indicator of high-risk fertility behavior is clearly correlated to the age of women in their sample (as the indicator take the value one when the current age is greater than 34). However, without contesting the veracity of this result, there is clearly a bias in the impact of the women’s age on the probability of presenting a positive risk. What would be the incidence of the mother’s age on her probability risk among mothers aged 33 or less?

The manuscript is technically sound, however, as to the geographical distribution of risky behavior, it is difficult to know whether the authors are talking about the Northern region or the Eastern region.

The authors argue that:” the country has been devastated by successive wars and armed conflict since 1996 in its Eastern part. As a consequence, the health system has worsened, especially in the Eastern region of the country. This has certainly increased the prevalence of high-risk fertility behaviors in the country”. But their findings reveal that higher risk are mostly prevalent in the northern part of the country. Why are the high spots of fertility risk not in the Eastern region? Which factors do explain the high spots in the Northern part of the country

The literature that studies the relationship between political instability, wars and fertility decisions seems to disagree on the nature of the relationship between armed conflict and fertility choices. On one hand, we have a group of studies that believes that political instability and related violence, not to mention economic uncertainties, negatively affect social returns such as population growth, and so it will logically follow a decline in fertility decisions and therefore less predominant risky fertility behaviors ((Berrebi and Ostwald (2015) , Caldwell (2004); Agadjanian et al (2011), McGinn (2000)). On the other hand, we have authors who rather think that armed conflicts would be likely to reinforce fertility decisions (Urdal and Che (2013), McGinn et al (2011)…), and thus increase risky fertility behaviors .

This article is not precise enough at this level, since it is difficult to identify if it is in the first or the second group. Indeed, the text is a bit ambiguous. It seems to indicate a strong prevalence of risky fertility behaviors in the North of the country.

This article would therefore stand out from the existing literature by identifying what is happening in the case of the Republic Democratic of Congo in the Eastern region of the country which has been affected by the war. And even more by identifying the mechanisms Through which the violence experienced by this region would reinforce the risky fertility behaviors. Highlighting such mechanisms could indeed be interesting targets for policy makers.

References:

Caldwell JC (2004). Social Upheaval and Fertility Decline. Journal of Family History, 29(4), 384–406

Agadjanian V, Yabiku T, & Boaventura C (2011). Men’s Migration and Women’s Fertility in Rural Mozambique. Demography, 48(3), 1029–48.

McGinn T (2000). Reproductive Health of War-Affected Populations: What Do We Know? International Family Planning Perspectives, 26(4), 174–18

Berrebi C & Ostwald J (2015). Terrorism and Fertility: Evidence for a Causal influence of Terrorism on Fertility. Oxford Economic Papers, 67(1), 63–82

McGinn T, Austin J, Anfinson K, Amsalu R, Casey SE, Fadulalmula S, & Yetter M (2011). Family Planning in Conflict: Results of Cross-Sectional Baseline Surveys in Three African Countries. Conflict and Health, 5(1), 11. H

Urdal H & Che CP (2013). War and Gender inequalities in Health: The Impact of Armed Conflict on Fertility and Maternal Mortality. International Interactions, 39(4), 489–510

Reviewer #3: This manuscript focuses on the correlates of what the authors call “high-risk fertility behaviour” (HRFB) in DR Congo. It shows that HRFB is frequent, and is linked to a variety of socioeconomic variables. Spatial variations and clustering are highlighted with various methods.

The manuscript is overall well written and presented and offers interesting findings. In my opinion, It has several serious shortcomings that should be addressed before being published.

The definition of high-risk fertility behaviour is not discussed in detail. The authors identify four criteria, but the reasons for selecting these criteria, and the choice for the thresholds are not discussed. For instance, the authors decide that having 4 at least four children puts the woman in a “high-risk fertility behaviour category”. Having a birth above age 34 is also considered “high-risk”. I think the authors should provide literature to support these choices. Considering these behaviours as high risk is also very much context-dependent, and this should be discussed.

The measurement is also debatable. As far as I understand (but I found it was not clear in the manuscript), young women (e.g. less than 35) are by definition less likely to be in the “high-risk” category, since they will not (yet) have had children above age 34. In the same vein, young women are less likely to be in the “high-risk category”, since they are less likely to have had 4 children. Overall, I found this indicator not to be specific (many dimensions are mixed), and very much related to the age of the women.

The identification of risk factors is not convincing. There are interesting results, but it looks like a series of variables was used in the regression model without theoretical justification. One finds some surprising results (e.g. for the standard of living), and the authors do not convincingly explain these results. In my opinion, this may be due to the fact that many variables are included in the model, some being endogenous. I do not understand why the variable one “decision to use contraception” is used in the model. The variable “ANC visits” is included. Does this mean that only ever-pregnant women are in the sample?

I could not figure out if the authors restricted their analyses to women in union. I suppose so since the partner’s level of education is used, but it is not clearly mentioned, and the consequence of this selection on the results is not discussed. N should be mentioned in Table 1.

Are community-level variables measured for primary sampling units?

Some surprising results deserve more discussion (partner’s level of education, household wealth index). Can this be due to collinearity (even if the VIF is not very high, this can be a substantive issue)?

I think the authors could improve the justification for testing interactions. I think they are not convincingly justified. Figures 1 and 2 could be dropped or moved to an appendix. Figure 3 is a nice figure but clearly shows that age is a major factor in high-risk fertility. This is not surprising given that having a birth above 35 is in the definition of HRFB. Figure 5 could be dropped. It looks like the output of a software package which is not discussed.

Spatial analyses offer interesting insights, but the description of the methods does not allow for understanding them in detail. In addition, I think the authors should select one or two approaches (e.g. the hot spots, or the interpolation). Figure 6 and Figure 7 are very interesting, but we expect some more discussion for these hotspots. In addition, what are the implications, from a programmatic point of view?

6. PLOS authors have the option to publish the peer review history of their article (what does this mean?). If published, this will include your full peer review and any attached files.

Reviewer #1: No

Reviewer #2: **Yes: **Bluette Arcady Mongoue

Reviewer #3: No

---

## [Decision Letter · Decision Letter 1]

13 Feb 2023

PONE-D-22-12958R1High-risk fertility behaviours among women of reproductive ages in the Democratic Republic of the Congo: Prevalence, Correlates and Spatial distributionPLOS ONE

Dear Dr. Tsala Dimbuene,

Thank you for submitting your manuscript to PLOS ONE. After careful consideration, we feel that it has merit but does not fully meet PLOS ONE’s publication criteria as it currently stands. Therefore, we invite you to submit a revised version of the manuscript that addresses the points raised during the review process.

We look forward to receiving your revised manuscript.

Kind regards,

Oyelola A. Adegboye, PhD

Academic Editor

PLOS ONE

Additional Editor Comments:

Although all reviewers agreed that this version of your manuscript has greatly improved, review 2 raised a crucial issue. You did not adequately address their comments. Critically address all comments and acknowledge the limitations of the study.

In addition to a point-by-point response, track or highlight all changes to your revised manuscript.

Reviewers' comments:

Reviewer's Responses to Questions

**Comments to the Author**

1. If the authors have adequately addressed your comments raised in a previous round of review and you feel that this manuscript is now acceptable for publication, you may indicate that here to bypass the “Comments to the Author” section, enter your conflict of interest statement in the “Confidential to Editor” section, and submit your "Accept" recommendation.

Reviewer #2: All comments have been addressed

Reviewer #3: (No Response)

Reviewer #4: All comments have been addressed

2. Is the manuscript technically sound, and do the data support the conclusions?

Reviewer #2: Yes

Reviewer #3: Partly

Reviewer #4: Yes

3. Has the statistical analysis been performed appropriately and rigorously? 

Reviewer #2: Yes

Reviewer #3: No

Reviewer #4: Yes

4. Have the authors made all data underlying the findings in their manuscript fully available?

Reviewer #2: Yes

Reviewer #3: Yes

Reviewer #4: Yes

5. Is the manuscript presented in an intelligible fashion and written in standard English?

Reviewer #2: Yes

Reviewer #3: Yes

Reviewer #4: Yes

6. Review Comments to the Author

Reviewer #2: (No Response)

Reviewer #3: Dear authors. Thank your your replies. You clarified some of the issues that were raised, but I find several were not adressed, or in a very brief way.

I think the restriction of the sample to "married women with at least one live birth" could be better justifed. Why don't you use women who have never had a live birth? With your approach, most women will be in the high risk category, unless they are between 19 and 34.

Why is age included in this way in the models (Table 2)? By defnition, women less that 19 and over 34 will be in the high risk category? The effect of age on the log odds is thus by definition not linear. Could you include age as a categorical variable?

How can you explain the "positive" effect of wealth on high risk fertility behaviour, especially in model 1? You mention "This finding might reflect the associations observed between husband/partner’s education and high-risk fertility behaviours". I do not understand what you mean.

Most previous comments still hold (Figure 5 not useful, not much justification for the variables, etc.).

Reviewer #4: The opening statement in the abstract about SDG is not necessary as there is no discussion about it afterwards.

The statement about "culture and social development" in the abstract is not necessary since they do not represent any unit of analysis in this study.

Line 112 says "theoretically": it is hard to know what it means in this context as there is no reference to any theory. Maybe the author mean studies. Check.

The study limitations should be expanded to include "non-interrogation of the possible contribution of armed

conflict" and "age of the respondents (women's).

Aside this, the authors have adequately addressed the reviewers' comments

7. PLOS authors have the option to publish the peer review history of their article (what does this mean?). If published, this will include your full peer review and any attached files.

Reviewer #2: No

Reviewer #3: No

Reviewer #4: **Yes: **Jimoh Amzat

---

## [Author Response · Author response to Decision Letter 1]

25 Feb 2023

Responses to reviewers have been provided in a point-by-point responses (see attached)

---

## [Editor Report · Decision Letter 2]

6 Mar 2023

High-risk fertility behaviours among women of reproductive ages in the Democratic Republic of the Congo: Prevalence, Correlates and Spatial distribution

PONE-D-22-12958R2

Dear Dr. Tsala Dimbuene,

We’re pleased to inform you that your manuscript has been judged scientifically suitable for publication and will be formally accepted for publication once it meets all outstanding technical requirements.

Kind regards,

Oyelola A. Adegboye, PhD

Academic Editor

PLOS ONE
---

## [Editor Report · Acceptance letter]

9 Mar 2023

PONE-D-22-12958R2 

High-risk fertility behaviours among women of reproductive ages in the Democratic Republic of the Congo: prevalence, correlates, and spatial distribution 

Dear Dr. Tsala Dimbuene:

I'm pleased to inform you that your manuscript has been deemed suitable for publication in PLOS ONE. Congratulations! Your manuscript is now with our production department. 

Kind regards, 

on behalf of

Dr. Oyelola A. Adegboye 

Academic Editor

PLOS ONE